

# Research on the application of biomaterial-based responsive hydrogels in the tumor microenvironment

Wanrong Yue[1,*], Xiaoyu Zhu[2,*], Ming Wu[3], Wenyue Qiang[4], Yixin Yang[5], Ziyun Zhang[6], Youwei Wang[7], Yuanyin Teng[8] and Mi Zhou[9]

[1] Department of Pathology, People's Hospital of Guilin, Guilin, Guangxi, China
[2] The Third Clinical Medical College, Ningxia Medical College, Yinchuan, Ningxia, China
[3] Orthopedics Department, Gongli Hospital, Shanghai, China
[4] Shanghai Jiao Tong University School of Medicine, Shanghai, China
[5] Department of Clinical Medicine, The First Clinical Medical College of Norman Bethune University of Medical Sciences, Changchun, Jilin, China
[6] School of International Education, Hainan Medical University, Haikou, Hainan, China
[7] Beihua University School of Medicine, Beihua University, Jilin, Jilin, China
[8] Institute of Hematology, Zhejiang University, Hangzhou, Zhejiang, China
[9] Department of Orthopedics, Tianjin Medical University General Hospital, Tianjin Medical University, Tianjin, China

[*] These authors contributed equally to this work.

Corresponding author
Mi Zhou, mizhou0524@tmu.edu.cn

## ABSTRACT

The tumor microenvironment (TME) is a critical factor influencing the initiation, growth, and spread of solid tumors. Recent advancements highlight biomaterial-based responsive hydrogels as promising smart materials within the TME owing to their high biocompatibility, biodegradability, and sensitivity to various stimuli. This review details the fundamental properties, preparation techniques, and diverse applications of responsive hydrogels, specifically within the TME. Key applications include their use in targeted tumor therapy, controlled drug delivery, and modulation of the complex TME. Recent studies have shown how these hydrogels, by responding to environmental factors such as pH, temperature, or specific biomolecules, can enhance treatment efficacy while minimizing systemic toxicity. Additionally, challenges are identified in terms of hydrogel design, safety, and clinical translation, with future directions aimed at optimizing hydrogel interactions with tumor-specific conditions. This review provides a comprehensive assessment of the current landscape, shedding light on the potential and ongoing research in the field of hydrogel-mediated cancer therapy.

## INTRODUCTION

Solid tumors represent a heterogeneous group of neoplasms originating from various tissues, significantly impacting patient health outcomes. The tumor microenvironment (TME), comprising extracellular matrix components, stromal cells, and immune cells, plays a crucial role in tumor progression, metastasis, and therapeutic response. This microenvironment not only facilitates tumor growth but also contributes to resistance

against conventional treatments, highlighting its importance in oncology (*Blanco & Wolfgang, 2019*). Traditional treatment modalities, including surgery, chemotherapy, and radiation, often encounter limitations due to systemic toxicity and inadequate targeting of the TME. Consequently, these approaches can result in adverse side effects and may fail to adequately address tumor heterogeneity, leading to suboptimal therapeutic outcomes (*Li et al., 2020*).

In recent years, there has been a growing interest in developing innovative therapeutic strategies that more effectively target solid tumors and their microenvironments. One promising approach involves the use of responsive hydrogels, which are three-dimensional polymer networks capable of undergoing reversible changes in response to external stimuli such as pH, temperature, or specific biomolecules (*Oliva et al., 2017*; *Zhu et al., 2022*). Hydrogels have evolved through three generations: first-generation hydrogels include (1) polymers derived from olefin monomers, (2) covalently cross-linked polymers, and (3) cellulose-based hydrogels for drug delivery; second-generation hydrogels feature PEG/polyester copolymers and stimulus-responsive hydrogels; third-generation hydrogels emphasize advanced cross-linking methods (*Cao et al., 2021*). Hydrogels currently serve multiple purposes in biomedical applications, including their use as medical wound dressings (*Liang, He & Guo, 2021*). Hydrogel-based dressings, characterized by their excellent flexibility and biocompatibility, effectively absorb fluids, thereby creating a moist environment conducive to tissue regeneration (*Amalakanti, Mulpuri & Avula, 2024*). In the realm of drug delivery, hydrogels exhibit significant potential due to their advantages in drug storage, controlled release rates, and triggerable release mechanisms (*Li & Mooney, 2016*). In tissue engineering—particularly in cardiac (*El-Sherbiny & Yacoub, 2013*), nerve (*Zhong et al., 2024*), and bone tissue repair (*Wang et al., 2025*),—hydrogels effectively mimic the extracellular-matrix environment, serving as carriers for cell transplantation; they enhance cell survival, proliferation, differentiation, and migration, thereby facilitating tissue regeneration. Furthermore, stimulus-responsive hydrogels, which integrate the advantages of standard hydrogels, can detect physical and chemical stimuli such as temperature, light, pH, and magnetic fields, leading to changes in their three-dimensional shape and phase state, which further promotes tissue repair (*Bhaladhare & Bhattacharjee, 2023*; *Chacón, Irineo-Moreno & Loera-Valencia, 2025*). The capacity of responsive hydrogels to encapsulate therapeutic agents (*Zhou et al., 2022*) and release them in a controlled manner renders them attractive options for enhancing the efficacy of cancer treatments while minimizing side effects (*Ma et al., 2024*).

The objective of this review is to explore the current understanding of solid tumors and their microenvironments, the limitations of traditional treatment methods, and the potential of responsive hydrogels as novel therapeutic strategies in oncology. By synthesizing recent advancements in this field, we aim to highlight the importance of integrating innovative materials into cancer therapy to improve patient outcomes and provide insights for future research directions.

## Rationale for the study

Cancer treatment remains a significant challenge due to the complexity of tumor biology and the heterogeneity of patient responses. The TME plays a crucial role in tumor progression and resistance to therapy, emphasizing the need for innovative approaches. Responsive hydrogels present a promising solution by enabling the controlled-release of therapeutic agents in response to TME stimuli, thereby enhancing localized treatment efficacy and improving patients' quality of life. However, research on responsive hydrogels is fragmented and yields varying efficacy and safety outcomes. While some studies report positive effects on tumor regression, others raise concerns regarding biocompatibility. The integration of these hydrogels with other therapies, such as immunotherapy, may address the limitations associated with single-agent treatments. This review aims to examine the properties and applications of responsive hydrogels within the TME, underscoring their potential to enhance cancer therapy and guide future research directions for improved patient outcomes.

## Description of the intended audience

The intended audience for this manuscript includes researchers and academics in the fields of oncology, biomedical engineering, and materials science, particularly those interested in innovative therapeutic strategies for cancer treatment. Additionally, it is relevant for clinicians and healthcare professionals seeking to enhance their understanding of the TME and the applications of responsive hydrogels in targeted drug delivery. Furthermore, graduate students and early-career scientists may find this review valuable for gaining insights into current trends and future directions in hydrogel-mediated cancer therapy. By addressing these diverse groups, the manuscript aims to foster interdisciplinary collaboration and advance knowledge in the development of effective cancer treatments.

## SEARCH METHODOLOGY SECTION

To comprehensively understand the application of hydrogels in the treatment of solid tumors by regulating the TME, we used keywords such as "responsive hydrogels", "TME", "cancer therapy", "hydrogel scaffold", "injectable hydrogel", "thermosensitive hydrogel", "supramolecular hydrogel", *etc*. We searched databases such as PubMed, EmBase, and Web of Science. The retrieval time limit was from January 2020 to January 2025.

The inclusion criteria for the literature were as follows: (1) related to the theme of hydrogels; (2) related to the clinical and basic research of tumor treatment; and (3) original articles and reviews. The exclusion criteria for the literature were as follows: (1) inability to obtain the full text and (2) the content being repetitive or similar. Our search strategy involved combining the keywords "AND" or "OR." Initial exclusion on the basis of titles and abstracts was performed independently by two reviewers (WR.Y. and M.Z.), followed by a further evaluation of full-text papers for eligibility.

# BASIC CHARACTERISTICS OF THE RESPONSIVE HYDROGELS

## Physicochemical properties

Responsive hydrogels are unique materials characterized by their ability to undergo significant changes in their physical and chemical properties in response to external stimuli such as temperature, pH, light, and ionic strength. These hydrogels typically exhibit a three-dimensional network structure composed of hydrophilic polymer chains that can swell or shrink depending on the environmental conditions. The physicochemical properties of these hydrogels are critical in determining their performance in various applications, particularly in biomedicine and drug delivery. For example, the swelling behavior of hydrogels can be finely tuned by modifying the polymer composition and crosslinking density, which directly affects their mechanical strength and elasticity. Recent studies have highlighted the potential of natural polymer-based hydrogels, which not only offer biocompatibility but also exhibit favorable mechanical properties suitable for various biomedical applications (*Jiang et al., 2020b*). Additionally, the incorporation of nanoparticles into the hydrogel matrix has been shown to increase their mechanical strength and responsiveness, paving the way for innovative applications in soft robotics and smart devices (*Yang et al., 2024*).

## Biocompatibility and biodegradability

The biocompatibility and biodegradability of responsive hydrogels are paramount for their application in medicine, particularly in drug delivery and tissue engineering. Biocompatible hydrogels are specifically designed to minimize adverse reactions upon contact with biological tissues, thereby ensuring safe and effective therapeutic outcomes. For instance, certain natural polymer materials, such as alginate and gelatin, promote cell adhesion and support cell growth on or within their surfaces due to their chemical compositions and structures being analogous to those of the extracellular matrix. Currently, novel injectable hydrogels, upon introduction into the human body, either do not trigger or only mildly trigger immune rejection reactions (*Li, Rodrigues & Tomás, 2012*; *Sun et al., 2020*). These hydrogels do not excessively activate the host immune system to attack the implanted biomaterials. Specifically, when in contact with blood, they do not elicit significant adverse reactions, such as hemolysis or coagulation, nor are they phagocytosed by immune cells (*Deng et al., 2021*). Furthermore, the biodegradability of these materials is crucial in applications where long-term implantation is undesirable. Responsive hydrogels can be engineered to degrade in response to specific biological stimuli, facilitating the controlled release of therapeutic agents and minimizing the need for surgical removal (*Pu et al., 2024*). Several key characteristics must be considered when designing the biodegradability features of responsive hydrogels. First, a controllable degradation rate is essential; for some large solid tumors, this type of hydrogel must maintain its structural integrity and functionality for a specified duration, which can be achieved by adjusting the degree of cross-linking and the length of the polymer chains (*Norouzi, Nazari & Miller, 2016*). Subsequent responsiveness to degradation is of utmost importance, meaning that in specific environments within tumor tissues, the hydrogel's degradation should be accelerated to
release drugs more precisely at the tumor site. Finally, the hydrogel must be completely metabolized in the body to prevent inflammatory and immune responses caused by long-term retention. Moreover, an important consideration in designing these methods is the evaluation of their metabolites and understanding their metabolic pathways (*Li, Yang & Lee, 2022*). Ongoing research continues to explore the mechanisms of biodegradation, emphasizing the role of enzymes and environmental conditions in the degradation process, which is essential for developing sustainable and effective biomaterials (*Jang et al., 2023*).

## Tunability and response mechanisms

A crucial factor to consider from the onset of developing or designing hydrogel materials is their ability to respond to changes in the TME (Fig. 1). These changes may include variations in temperature, light, pH, redox potential, magnetic fields, hardness, or the presence of specific chemical molecules. Consequently, this aspect is paramount when addressing the diverse characteristics of TMEs. Specifically, the responsive properties of a material must align with the inherent traits of the tumor itself. Essentially, these stimulus-responsive behaviors are governed by the interactions among polymers and between polymers and solvents (*Sun et al., 2020*). The tunability of responsive hydrogels is one of their most appealing features, enabling researchers to design materials that respond to specific stimuli with precision. This tunability is achieved through manipulation of polymer chemistry, cross-linking density, and the incorporation of various functional groups. For instance, thermoresponsive hydrogels can be engineered to alter their swelling behavior at designated temperatures, rendering them ideal for applications in drug-delivery systems that necessitate controlled release (*Qian et al., 2023*). Furthermore, the response mechanisms of these hydrogels can be augmented by integrating photothermal agents, allowing them to respond to light stimuli (*Pu et al., 2024*). Additionally, thermosensitive hydrogels are among the most prevalent stimuli-sensitive hydrogels developed in recent years. Some researchers have developed chitosan-based hydrogels for the treatment of colon cancer. In *in vivo* experiments involving intratumoral injection, hyperthermia can be combined with chemotherapy. In experiments conducted on tumor-bearing mice, this system has demonstrated significant antitumor effects and can achieve continuous release of Dox and MBP at body temperature (*Zheng et al., 2019*). *Zhao et al. (2019)* developed an *in situ* injectable hydrogel based on a photopolymerizable hydrogel to co-deliver paclitaxel and temozolomide, aiming to prevent glioblastoma recurrence following surgical resection. This hydrogel delivery system for glioblastoma multiforme (GBM) is sterile, injectable, adheres to the resection cavity, and is biocompatible (*Zhao et al., 2019*). The primary factor in designing pH-sensitive hydrogels is that the pH of the TME is acidic (ranging from 6.0 to 7.1) (*Maimela, Liu & Zhang, 2019*), and endosomes and lysosomes have an even more acidic environment (*Worsley, Veale & Mayne, 2022*; *Zhang, Lin & Gillies, 2010*). Compared with normal physiological pH, these acidic conditions provide an environmental cue for the degradation of responsive hydrogels and drug release. The specific response mechanism involves cleavable linkers utilized for pH responsiveness. The protonation of certain ionizable groups (such as carboxyl groups) or acidic cleavable bonds can lead to cleavage when the pH changes. *Raza et al. (2019)* developed a pH-sensitive hydrogel

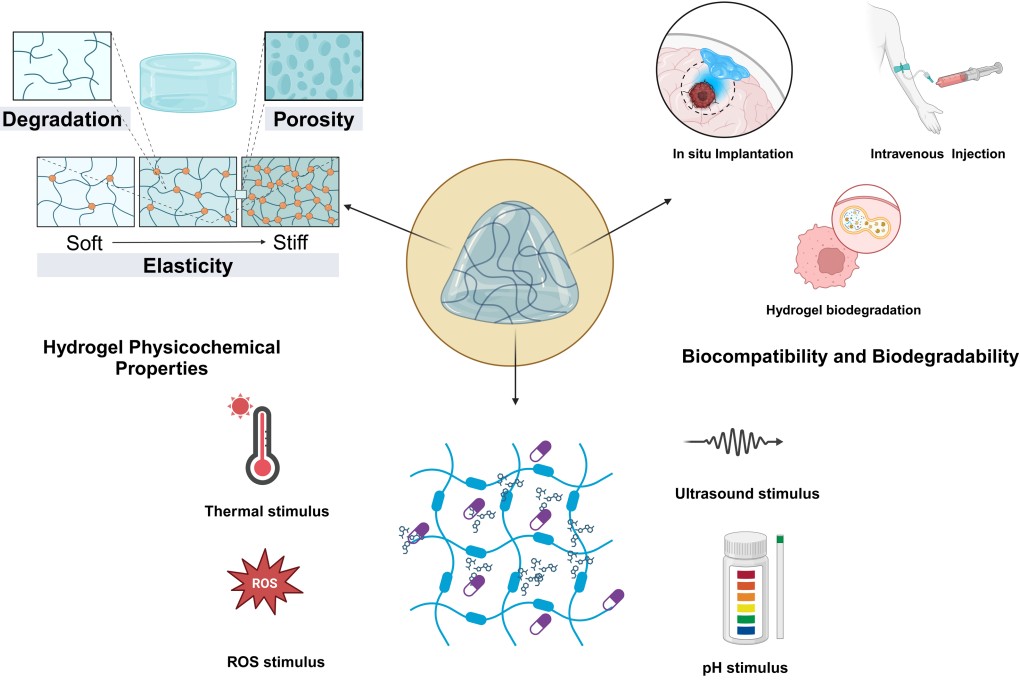

**Figure 1** **Basic characteristics of responsive hydrogels.** The physicochemical properties, biocompatibility and biodegradability, as well as the tunability and responsive mechanisms of the hydrogels.

based on the fer-8 peptide hydrogel. In a liver-cancer model, following intratumoral injection, it can detect pH changes and sustainably release paclitaxel. In tumor tissues, the concentration of glutathione (GSH) is relatively high, with levels in the cytoplasm being 1,000 times greater than those in healthy tissues and significantly higher than in the extracellular matrix (*Gamcsik et al., 2012*; *Wang et al., 2022a*). Therefore, redox-responsive hydrogels are typically composed of chemical bonds that degrade in the presence of high concentrations of intracellular GSH, such as disulfide and diselenide bonds. Building on this strategy, *Zhang et al. (2017)* developed a redox-responsive hydrogel carrying doxorubicin based on a dextrin hydrogel. After intravenous administration, it can precisely target breast-cancer lesions and inhibit tumor progression and metastasis (*Zhang et al., 2017*). The development of multistimuli-responsive hydrogels has opened new avenues for creating smart materials that can adapt to varying tumor microenvironmental conditions (*Khattak et al., 2025*), thus enhancing their *in situ* applicability. Moving forward, to develop more responsive and controllable hydrogels, it is essential to simulate and study the release kinetics of hydrogels and their triggering conditions more precisely in diverse environments (*Chen et al., 2021*; *Zhou et al., 2023*). As research advances, a deeper understanding of the underlying mechanisms governing these responses will facilitate more sophisticated designs and applications of responsive hydrogels in biomedicine and beyond (*Cao et al., 2023*).

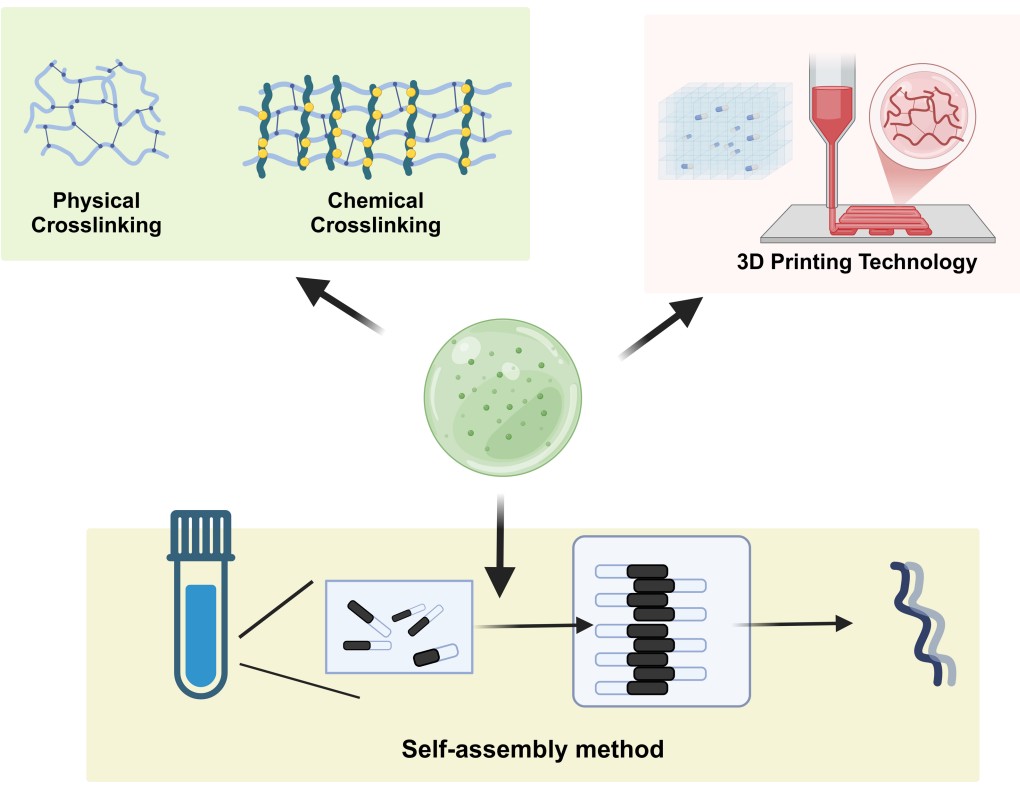

**Figure 2** **Synthesis methods of responsive hydrogels.** This demonstrates the self-assembly of hydrogels, chemical crosslinking, physical crosslinking, and the 3D-printing strategy.

# SYNTHESIS METHODS FOR THE RESPONSIVE HYDROGELS

## Self-assembly method

Self-assembly is a prominent technique for the preparation of responsive hydrogels and is characterized by the spontaneous organization of molecules into well-defined structures without external guidance (Fig. 2). This method leverages the intrinsic properties of the materials, such as hydrogen bonding, hydrophobic interactions, and van der Waals forces, to drive the assembly process. For example, bioinspired hydrogels utilize self-assembly to create complex architectures that mimic biological tissues. Self-organization can be influenced by various factors, including pH, temperature, and ionic strength, allowing the design of hydrogels with tunable properties and functionalities. Recent advancements have demonstrated the potential of self-assembled hydrogels in drug delivery systems, where their responsiveness to environmental stimuli enhances the controlled release of therapeutic agents, thus improving treatment efficacy and minimizing side effects (*Rigby et al., 2023*; *Wei et al., 2024*). Additionally, the use of self-assembly to create layered structures has shown promise in the development of scaffolds for tissue engineering applications, where hierarchical organization can support cell growth and differentiation (*Rigby et al., 2023*; *Zhang et al., 2022*).

## Chemical crosslinking method

Chemical crosslinking is a widely used method for synthesizing hydrogels and involves the formation of covalent bonds between polymer chains, which enhances the mechanical stability and durability of the resulting materials (*George et al., 2019*; *Gila-Vilchez et al., 2022*). This method can employ various crosslinking agents, such as glutaraldehyde, epoxides, or multifunctional monomers, to create networks that can respond to environmental changes. The degree of crosslinking can be precisely controlled, allowing for the customization of hydrogel properties, such as swelling behavior and mechanical strength. For example, hybrid crosslinking strategies that combine both covalent and noncovalent interactions have been explored to produce hydrogels with superior mechanical properties and self-healing capabilities, making them suitable for biomedical applications (*Jiang et al., 2023*; *Zhang et al., 2021a*; *Zheng et al., 2022a*). Moreover, chemically crosslinked hydrogels have been utilized in drug delivery systems, where the release profile of encapsulated drugs can be modulated by adjusting the crosslinking density, thereby enhancing therapeutic outcomes (*Jiang et al., 2020a*; *Luo et al., 2023*).

## Physical crosslinking method

Physical crosslinking involves the formation of hydrogels through noncovalent interactions, such as hydrogen bonding, ionic interactions, and hydrophobic effects (*Hu et al., 2019*). This method offers several advantages, including mild reaction conditions and the ability to create reversible networks, which can be beneficial for applications requiring dynamic properties. For example, ionic gelation, a common physical crosslinking technique, utilizes the interaction between oppositely charged polymers to form hydrogels. This method has been effectively applied in the development of alginate-based hydrogels, which are widely used in drug delivery and tissue engineering because of their biocompatibility and biodegradability (*Hu et al., 2023*; *Shamma et al., 2022*). Additionally, physical crosslinking can be employed to create smart hydrogels that respond to external stimuli, such as temperature and pH, enabling their use in controlled drug release systems and biosensors (*Farokhi et al., 2021*; *Maiti, Imani & Yoon, 2021*).

## Application of 3D printing technology in hydrogel preparation

3D printing technology has revolutionized the fabrication of hydrogels, allowing for precise control over their structure and composition at the microscale (*Barcena et al., 2023*; *Zhong et al., 2023*). This method enables the creation of complex geometries and customized scaffolds that can mimic the extracellular matrix, providing a supportive environment for cell growth and tissue regeneration. Various 3D printing techniques, including extrusion-based printing and bioprinting, have been utilized to fabricate hydrogels with tailored mechanical and biochemical properties (*Zhou et al., 2020*). For instance, bioinks composed of natural polymers such as gelatin and alginate have demonstrated significant potential in tissue engineering applications, where printed scaffolds facilitate cell adhesion and proliferation (*Liu, Tagami & Ozeki, 2020*; *Mallakpour, Tukhani & Hussain, 2021*). Furthermore, advancements in 3D printing have enabled the development of drug-delivery systems capable of releasing therapeutic agents in a controlled manner, thereby enhancing

treatment efficacy (*Hu et al., 2023*; *Shamma et al., 2022*). Overall, the integration of 3D printing technology in hydrogel preparation signifies a substantial advancement in the field, paving the way for new biomedical applications.

## APPLICATIONS OF RESPONSIVE HYDROGELS IN THE TME

### Drug delivery systems

Owing to the limitations of traditional drug delivery methods, hydrogel-based delivery systems have drawn significant attention as novel carriers for tumor drugs (*Li & Mooney, 2016*). Hydrogels are composed of a large amount of water and cross-linked polymer networks, featuring excellent biocompatibility and the ability to protect drugs (*Caló & Khutoryanskiy, 2015*; *Li & Mooney, 2016*; *Trinadha Rao et al., 2021*). Their multiscale properties facilitate the controlled delivery of drugs. On the basis of size, hydrogels can be classified into macroscopic hydrogels, microgels (0.5–10 µm), and nanogels (<200 nm) (*Karg et al., 2019*). Macroscopic hydrogels are often implanted surgically or used for transepithelial drug delivery. Owing to their small size, microgels and nanogels are suitable for various routes of administration, such as intravenous injection and *in situ* injection.

The mesh size of hydrogels directly impacts drug diffusion and release. The drug release rate can be regulated by controlling network degradation, swelling, and mechanical deformation. Additionally, through covalent bonding, electrostatic interactions, and hydrophobic interactions, drugs can be released slowly or on demand. Certainly, different drug administration routes have their own advantages and disadvantages. For example, researchers have reported that intravenous injection (*Wilhelm et al., 2016*), pulmonary delivery (*Zhu et al., 2017*), and oral delivery (*Lin et al., 2015*) offer high absorption rates. However, drugs administered *via* these routes are prone to rapid clearance by the liver and kidneys. *In situ* injection and implantation, on the other hand, can avoid the first-pass elimination effect. Nevertheless, they inevitably require surgical procedures for transplantation. Currently, the microneedle patch, as a novel drug delivery method, has advantages over other hydrogels. It features convenient administration, high permeability, low toxicity and side effects, avoidance of first-pass liver metabolism, and a reduction in pain and discomfort (*Ganeson et al., 2023*; *Seetharam et al., 2020*). In the treatment of superficial tumors such as melanoma, microneedles can penetrate the epidermis painlessly, enhance drug delivery, and continuously and specifically induce an antitumor immune response in the lymph nodes (*Wang et al., 2016*). Studies have demonstrated that hydrogel-based systems can significantly improve the pharmacokinetics and bioavailability (*Xie et al., 2021*) of anticancer drugs, leading to enhanced tumor targeting and reduced toxicity to healthy tissues. Furthermore, the incorporation of targeting ligands within these hydrogels can facilitate selective binding to cancer cells, further increasing the precision of drug delivery. The versatility of responsive hydrogels in drug delivery applications represents a promising avenue for improving cancer treatment outcomes (Fig. 3).

### Targeting action on tumor cells

The ability of responsive hydrogels to target tumor cells is crucial for cancer therapy. By functionalizing hydrogels with ligands or antibodies that recognize overexpressed receptors

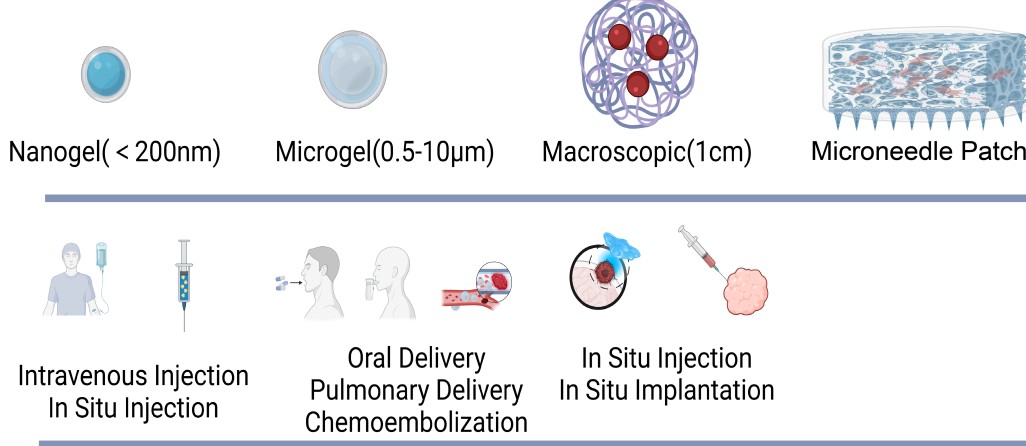

**Figure 3  Hydrogel delivery system and routes.** The various hydrogel delivery systems, categorized by their size (nanogel, microgel, macroscopic, microneedle patch), and their corresponding delivery routes, which vary accordingly.

on cancer cells (*Prajapati et al., 2024*), these materials enable targeted delivery of therapeutic agents. This approach increases drug concentration at the tumor site while minimizing off-target effects common in conventional chemotherapy. Recent advancements have demonstrated hydrogels that respond to tumor-specific biomarkers, allowing selective drug release in their presence (*Jain et al., 2024*; *Wang et al., 2022b*). Furthermore, the dynamic nature of these hydrogels allows them to adapt to the tumor's changing environment, potentially addressing challenges like drug resistance and heterogeneity within the TME (*Lin, Zhao & Chen, 2024*; *Xue et al., 2018*). This targeted action enhances the therapeutic index of anticancer agents, making responsive hydrogels valuable tools in personalized cancer-treatment strategies.

## Microenvironment regulation and tumor growth inhibition

Responsive hydrogels play a crucial role in modulating the TME, which is essential for inhibiting tumor growth. The TME involves a complex interaction of cells, extracellular matrix components, and signaling molecules that can promote tumor progression and metastasis. By incorporating bioactive agents into hydrogels, researchers can alter this microenvironment to suppress tumor growth. For instance, hydrogels can be engineered to release anti-inflammatory or immunomodulatory agents that modify the immune landscape of tumors, potentially enhancing immunotherapy effectiveness (*Yu et al., 2022*). Additionally, hydrogels can deliver agents targeting the stroma or vascular components of tumors (*Xu et al., 2022*), disrupting the supportive environment necessary for tumor survival and growth (*Jiang et al., 2024*). This dual strategy of delivering therapeutic agents

while modifying the TME offers a promising approach to improve cancer treatment outcomes and may lead to more effective therapies in the future (*Xing et al., 2022*).

## LATEST RESEARCH FINDINGS

### Recent preclinical studies and trials

In the field of cancer and regenerative medicine, significant progress has been made in pre-clinical research, providing new ideas for disease treatment. The psilin research conducted using zebrafish as the experimental model fully demonstrated the advantages of this animal model in revealing mechanisms of drug action (*Syed, Tsang & Gerlai, 2023*). Meanwhile, the breakthrough development of xenotransplantation technology has effectively alleviated the problem of insufficient organ donors and opened an innovative path for transplant medicine (*Lu et al., 2019*). It is notable that incorporating gender factors as biological variables in pre-clinical studies has significantly improved the accuracy of evaluating therapeutic interventions across populations (*Justice, 2024*). These breakthroughs not only deepen our understanding of disease mechanisms and treatment responses, but also provide a solid theoretical basis for clinical practice.

In cancer treatment, hydrogel applications have entered clinical trials. As an important modality for malignant tumours, pelvic- and abdominal-radiotherapy usage ranges from 20% to 60% (*Hessels et al., 2022*; *Viswanathan et al., 2014*). However, radiation injury remains difficult to avoid: the rectum, sigmoid colon, pelvic small intestine, and ileocecal zone are most vulnerable, with radiation proctitis showing the highest incidence and longest duration. To address this challenge, researchers developed spacer-hydrogel systems. Implanted between the tumour and rectum, they reduce rectal dose and injury risk. SpaceOAR hydrogel, the first PEG-based product with FDA clearance, has been pivotal in prostate-cancer radioprotection (*Mariados et al., 2015*; *Miller et al., 2020*). Long-term follow-up shows that patients receiving hydrogel injections have improved bowel, urinary, and sexual function, and a reduced risk of secondary malignancies (*Grewal et al., 2023*; *Karsh et al., 2018*).

The combination of recombinant human bone morphogenetic protein-2 (rhBMP-2) and an absorbable collagen-sponge carrier induces bone formation and can replace autologous bone in certain interbody spinal fusions (*McKay, Peckham & Badura, 2007*). Because BMPs and their receptors are expressed in tumours, some scholars have questioned cancer risk. A systematic review reported cancer incidences of 12.5% and 5.6% in rhBMP-7 groups *versus* 8.3% and 0% in controls, differences that were not statistically significant (*Devine et al., 2012*).

For poor-prognosis colorectal cancer (CRC), a comparative study is evaluating FOLFOX6m plus monoclonal-antibody (mAb) therapy *versus* the same regimen combined with liver chemo-embolisation *via* LifePearls-Irinotecan (*Paez et al., 2021*). A novel reverse thermosensitive hydrogel, UGN-102, has been developed for intravesical chemoablation for the treatment of recurrent low-grade nonmuscle invasive bladder cancer (*Prasad et al., 2025*). A novel reverse-thermosensitive hydrogel, UGN-102, has been developed for intravesical chemo-ablation of recurrent low-grade, non-muscle-invasive bladder cancer.

A phase-3 trial showed a 3-month complete-response rate of 65% for UGN-102 *versus* 64% for transurethral resection of bladder tumour (TURB). At 15 months, disease-free-survival probability was 72% for UGN-102, significantly greater than the 50% with TURB (*González-Padilla, Subiela & Villacampa-Aubá, 2024*). Moreover, LICORN-01—a hydrogel loaded with granulocyte-macrophage colony-stimulating factor (GM-CSF) and mifamurtide—is now in phase-I trials for colorectal liver metastases (*Wang et al., 2024b*).

## Combined application of hydrogels and immunotherapy

Over the past few decades, immunotherapy has fundamentally transformed cancer treatment. Broadly, current immunotherapeutic strategies fall into five categories: (1) immunomodulators, (2) immune-checkpoint-blocking mAbs such as PD-1/PD-L1 inhibitors, (3) therapeutic cancer vaccines, (4) adoptive-cellular therapies, and (5) oncolytic viruses. Immune-checkpoint inhibitors restore antitumour immunity by blocking checkpoint molecules expressed on tumour or immune cells, thereby lifting immunosuppression and enabling cytotoxic lymphocytes to recognise and destroy cancer cells. In contrast, cancer vaccines deliver tumour-associated antigens—either as whole proteins or defined peptide fragments—to prime and amplify antigen-specific T-cell responses against malignant cells.

The current research hotspots focus on the synergistic effect of hydrogel materials and immunotherapy technologies, aiming to enhance their application value in the field of tumour treatment. Multiple empirical studies have confirmed that this combined strategy can significantly improve clinical efficacy (*Erfani, Diaz & Doyle, 2023*). The core advantage of the hydrogel system lies in its region-specific drug-delivery ability, which can effectively reduce the toxicity and side effects caused by systemic administration. By regulating the molecular cross-linking network and physical structure of hydrogels, personalised drug-release patterns can be achieved to ensure that therapeutic agents maintain a stable, effective concentration in the lesion area. This sustained-release design not only reduces the frequency of drug administration, but also decreases the need for repeated pre-treatment tests, thereby optimising the patient experience. Experimental data confirm that local injection of hydrogels can precisely deliver therapeutic drugs—such as immune-checkpoint inhibitors—to the tumour microenvironment, significantly improving therapeutic effect (*Zhang et al., 2021b*). More importantly, the hydrogel matrix can reshape the TME, enhance T-cell infiltration, and effectively inhibit tumour immune-escape mechanisms (*Xu et al., 2024*). Studies have shown that by functionalising hydrogels, controllable release of therapeutic drugs can be achieved while exerting the dual effects of drug delivery and immune regulation (*Xie et al., 2021*). This novel therapeutic strategy not only enhances the targeting of tumour treatment, but also minimises systemic adverse reactions, fully demonstrating the application potential of hydrogels in tumour immunotherapy.

## Combination of biomaterials and gene therapy

In tumor therapy, gene therapy utilizing responsive hydrogels has high specificity, effectively overcoming the limitations of traditional tissue-engineering techniques and showing significant advantages in tissue repair. However, therapeutic genes are often hindered by

cellular barriers and enzymatic sensitivity, making the loading of gene carriers a critical factor. Gene-therapeutic hydrogels excel in addressing these challenges and have achieved remarkable progress in this domain (*Xie et al., 2021*). Common therapeutic genes include RNA, microRNAs, small plasmids, interference RNA fragments and exosomes (*Chao et al., 2025*); however, they are often prone to degradation and exhibit poor stability. Although targeted RNA delivery has potential for disease treatment, therapeutic genes are susceptible to degradation by ribonucleases and face significant barriers from both the extracellular and intracellular environment. *Zhao et al. (2016)* developed an injectable bioactive hydrogel system for melanoma treatment. This system functions by up-regulating tumor-suppressor genes and down-regulating oncogenes. In vivo studies have demonstrated that the hydrogel system can significantly inhibit tumor growth in both breast-cancer-bearing and melanoma-bearing mouse models. Additionally, it can transform the morphology of melanoma B16 cells into a melanin-producing phenotype. High-throughput sequencing further revealed that tumor-suppressor genes are up-regulated and oncogenes are down-regulated, confirming that the developed system can selectively activate certain tumor-suppressor genes and inactivate specific oncogenes, thus facilitating the benign reversion of the tumor phenotype. Similarly, to address the challenge of constructing delivery systems for small siRNAs or shRNAs *in vivo*, researchers have developed self-assembled nanogels based on cholesterol-modified cyclodextrin with spermine groups (CH-CA-Spe). This nanogel is loaded with vascular endothelial growth factor-specific small interfering RNA (siVEGF). It successfully inhibited angiogenesis and tumor growth in a renal cell carcinoma mouse model (*Fujii et al., 2014*). These findings validate the potential of the nanogel as a clinically viable drug-delivery system for the intratumoral delivery of therapeutic siRNA in the future. One notable area of progress is the use of biomaterials in clustered regularly interspaced short palindromic repeats/Cas9 (CRISPR/Cas9) delivery, where biomaterials can facilitate the precise editing of genes while minimizing off-target effects (*Dubey & Mostafavi, 2023*). To further translate and apply CRISPR/Cas9 technology in cancer gene therapy, efficient and highly specific delivery of the Cas9 protein and single-guide RNA to tumor sites is crucial (*Dogra et al., 2025*). *Chen et al. (2017)* developed a novel type of liposome-templated hydrogel nanoparticle (LHNP). In animal experiments, LHNPs achieved targeted inhibition of multiple genes in various tumors, including brain tumors, effectively suppressing tumor growth and improving the survival rate of tumor-bearing mice (*Chen et al., 2017*). Additionally, studies have shown that biomaterials can support the *in vivo* generation of chimeric antigen receptor T (CAR-T) cells, which are pivotal in immunotherapy for various cancers (*Qin et al., 2023*). However, in solid tumors, due to insufficient infiltration of CAR-T cells within the tumor, researchers have developed injectable supramolecular hydrogel systems to load a plasmid encoding CAR (pCAR) under the control of a T-cell-specific CD2 promoter. In humanized mouse models, this system has successfully achieved *in situ* generation and effective accumulation of CAR-T cells at the tumor site (*Zhu et al., 2024*). The intersection of biomaterials and gene therapy not only enhances the feasibility of non-viral delivery systems but also provides a platform for developing innovative treatments tailored to individual patient needs, highlighting the transformative potential of this research area.

# CHALLENGES AND FUTURE DIRECTIONS

## Complexity of design and preparation

The complexity of design and preparation in biomedical applications presents significant challenges that must be addressed to enhance efficacy and safety (*Su et al., 2024*). The intricacies involved in developing biomaterials, drug-delivery systems, and therapeutic agents often require interdisciplinary approaches that integrate materials science, biology, and engineering. For instance, designing multifunctional polyion complex vesicles necessitates a comprehensive understanding of polyelectrolyte interactions and their implications for drug encapsulation and release profiles (*Huang et al., 2022*). Moreover, the quality-by-design (QbD) approach has emerged as a promising strategy for the systematic development of controlled-release preparations, particularly in traditional medicine (*Wang et al., 2019*). These methodologies not only streamline the design process but also ensure that the final products meet rigorous safety and efficacy standards. However, the inherent complexity of these designs can lead to variability in performance, necessitating robust characterization techniques to evaluate their physical and chemical properties (*Azevedo & Mata, 2022*). As the field progresses, there is a pressing need for innovative design frameworks that can accommodate the multifaceted requirements of modern therapeutics while also being scalable for clinical applications.

## Safety and biocompatibility assessment

In the research and development process of biomedical materials and devices, the testing of material safety and biocompatibility is an indispensable link (*Seo et al., 2022*). When systematically evaluating the interaction between materials and biological systems, it is necessary to focus on the differences caused by their chemical composition and physical structural characteristics (*Street & Christian, 2024*; *Kavasi et al., 2021*). It is worth noting that for new materials such as biodegradable stents, strict long-term safety tests need to be carried out to prevent possible negative effects after implantation (*Dinu et al., 2022*; *Zheng et al., 2022b*). Facing an increasingly improved regulatory system, researchers must not only follow standardized testing procedures but also actively develop innovative evaluation methods, such as 3D *in vitro* models, to enhance the accuracy of biocompatibility testing (*Dinu et al., 2022*). This improvement will significantly accelerate the process of transforming new biomaterials from basic research to clinical application.

## Challenges in clinical translation

In the transformation process of biomedical technology from laboratory research to clinical application, multiple challenges persist, the key issue being the gap between experimental data and actual therapeutic effects—often referred to as the "Valley of Death" in translational medicine. Although hydrogels have performed well in pre-clinical studies of cancer immunotherapy, numerous obstacles remain before clinical application and final approval. The primary challenge lies in innovating hydrogels as drug carriers, which requires long-term research to verify their *in vivo* safety. Transformation challenges encompass material design, good manufacturing practice (GMP)-standardized production, and regulatory approval. In particular, the clinical translation of nanomedicine is difficult

because of complex factors such as regulatory-approval processes, production scale, and biocompatibility (*Metselaar & Lammers, 2020*). Meanwhile, the new therapy lacks unified standards for safety- and efficacy-evaluation, which may increase uncertainty and thus hinder clinical promotion (*Sigmund et al., 2020*). To address these challenges and optimize the transformation process, collaboration among researchers, clinical experts, and regulatory agencies is crucial (*Biomarkers of Aging Consortium et al., 2024*). Notably, the introduction of cutting-edge technologies such as artificial intelligence and machine learning facilitates biomarker identification and patient stratification, thereby promoting personalized treatment (*Wang et al., 2024a*). As the field progresses, solving practical problems in clinical applications while encouraging innovation will become key (*Lu et al., 2024*). The primary considerations in hydrogel development focus on biosafety characteristics and structural stability, requiring the material to have good biocompatibility to reduce host-immune rejection and prevent chronic inflammation. Meanwhile, the material must maintain its structural integrity and functionality in the *in vivo* environment. Furthermore, systematic research on the TME urgently needs to be deepened, with emphasis on detecting hydrogel response mechanisms to various biological stimuli and analyzing how different microenvironment parameters influence controlled-release behaviour. By integrating theories and methods from materials science, bioengineering, and clinical medicine, precise optimisation of manufacturing processes can be achieved. This not only enables the construction of more refined design plans but also provides reliable technical support for the implementation of precision medicine.

## CONCLUSION

This review deeply explores the key role of responsive hydrogels in the TME and their application prospects in the field of tumor treatment. As tumor-biology research continues to advance, these materials have demonstrated unique advantages. Responsive hydrogel systems can achieve precise targeting and local treatment by sensing specific micro-environmental signals, thereby enhancing therapeutic effects and reducing systemic adverse reactions. However, existing studies show inconsistencies in efficacy evaluation and safety analysis; therefore, a systematic evaluation system is urgently needed to reach scientific consensus. Notably, combining hydrogels with other treatment methods, such as immunotherapy or radiotherapy, can effectively break through the limitations of single therapies. Key research directions for the future include optimizing material formulations to enhance biocompatibility and degradation performance, thoroughly exploring their interaction mechanisms with the immune microenvironment, and developing advanced *in vivo* imaging technologies. Overall, responsive hydrogels have brought revolutionary breakthroughs to tumor treatment, and realizing their full potential will depend on multidisciplinary collaborative innovation.

### Funding

The authors received no funding for this work.

### Competing Interests

The authors declare there are no competing interests.

### Author Contributions

- Wanrong Yue conceived and designed the experiments, authored or reviewed drafts of the article, and approved the final draft.
- Xiaoyu Zhu conceived and designed the experiments, analyzed the data, authored or reviewed drafts of the article, and approved the final draft.
- Ming Wu analyzed the data, authored or reviewed drafts of the article, and approved the final draft.
- Wenyue Qiang performed the experiments, authored or reviewed drafts of the article, and approved the final draft.
- Yixin Yang performed the experiments, authored or reviewed drafts of the article, and approved the final draft.
- Ziyun Zhang performed the experiments, analyzed the data, authored or reviewed drafts of the article, and approved the final draft.
- Youwei Wang performed the experiments, analyzed the data, prepared figures and/or tables, authored or reviewed drafts of the article, and approved the final draft.
- Yuanyin Teng analyzed the data, authored or reviewed drafts of the article, and approved the final draft.
- Mi Zhou conceived and designed the experiments, performed the experiments, analyzed the data, prepared figures and/or tables, authored or reviewed drafts of the article, and approved the final draft.

### Data Availability

This is a literature review.

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
