# Peer review of "Research on the application of biomaterial-based responsive hydrogels in the tumor microenvironment"

_PeerJ, doi:10.7717/peerj.19609_

## Round 0.1 · original submission · Major Revisions

Dear authors,

Thank you for your submission. I think this work should be revised to better analyse current and future paradigms and indeed contribute to the current scientific literature on the subject. Please, refer to the reviewers' comments for further details.

Reviewer 1 ·

Basic reporting

The manuscript titled “Research on the Application of Biomaterial-Based Responsive Hydrogels in the Tumor Microenvironment” contains general information that has already been extensively reported in the literature. To be considered for publication, major revisions are required.

1. The manuscript lacks tables and figures, which are essential to provide readers with a clearer understanding of the content. Adding a table summarizing different examples of hydrogels for biomedical applications would improve the manuscript.
2. Another major drawback is the absence of specific examples from the literature. The authors should include at least 3–5 examples illustrating the applications of hydrogels in drug delivery and other applications. Additionally, the role of hydrogels in stem cell-based therapy, tissue engineering, and regenerative medicine should be thoroughly explained to enhance the manuscript's scientific significance.
3. The references cited are insufficient and not comprehensive. Additional references should be included, such as at lines 95–104 and lines 114–115, to support the discussion of natural polymers like chitosan-based hydrogels.
4. The conclusion should be revised to be short and concise, clearly summarizing the main findings and their implications.

Experimental design

No comments

Validity of the findings

No comments

Additional comments

No comments

Reviewer 2 ·

Basic reporting

The aim of the manuscript is about " Application of Biomaterial-Based Responsive Hydrogels in the Tumor Microenvironment". The disscussion should foucus on how do hydrogels response tumor microenvironment environmental factors, such as pH, temperature, or specific biomolecules, and who do responsive properties of hydrogels associate their applications. The structure of Main Body should be appropriately adjusted.

Experimental design

no comment

Validity of the findings

no comment

Additional comments

NO

Reviewer 3 ·

Basic reporting

This manuscript aims to investigate the current knowledge of solid tumors and their microenvironments, the limitations of conventional treatment approaches, and the potential of responsive hydrogels as an innovative therapeutic strategy in oncology, as outlined in the introduction. However, a dedicated section illustrating the tumor microenvironment is needed for better clarity.
Additionally, the first part of the introduction (sections 2.1 and 2.2) focuses on the fundamental characteristics of responsive hydrogels. While this constitutes nearly half of the manuscript, it includes repetitive findings that are not directly aligned with the core objective of the review. These sections should be more concise.
To enhance the manuscript’s clarity, consider including:
• A new figure summarizing the properties of hydrogels (section 2.1).
• A schematic representation of hydrogel preparation methods (section 2.2).
• A figure summarizing their applications in the tumor microenvironment (section 2.3).
• A table organizing the latest research findings (section 2.4).
These additions would improve the illustration and accessibility of the findings.

- Clarity & Language: The manuscript is written in clear and professional English. However, some sections could benefit from minor grammatical improvements and better flow to enhance readability.
- Context & Literature Review: Although the manuscript provides a well-referenced literature review, the sections 2.1 and 2.2 present repeated findings not related to the core objective of the review. The citations are appropriate and up-to-date. However, certain sections, particularly section 2.4, could include more recent studies to strengthen the discussion.
- Structure & Journal Standards: The article follows PeerJ standards and discipline norms. It maintains a logical structure of review article with a clear introduction, main body, and conclusion.
- Scope & Interest: The manuscript fits well within the journal's scope and is of broad interest to the biomaterials and cancer research community.

Experimental design

- Relevance to Journal Scope: The article aligns with the journal’s aims and scope, focusing on responsive hydrogels for cancer therapy.
- Methodology & Rigor: The review methodology is well-defined, incorporating a systematic literature search. However, the inclusion/exclusion criteria for selected references could be more explicitly detailed.
- Coverage & Organization: The review provides comprehensive coverage of responsive hydrogels in the tumor microenvironment. It is well-structured with coherent paragraphs and subsections. However, it would benefit from a clearer comparison of different hydrogel types and their specific advantages/disadvantages.

Validity of the findings

- Impact & Novelty: The manuscript does not claim novelty but provides a comprehensive synthesis of current research. The discussion effectively highlights gaps and future directions in hydrogel-mediated cancer therapy.
- Conclusions & Supporting Evidence: The conclusions are well-stated and supported by referenced literature. However, some claims (e.g., about the clinical translation of hydrogels) could be reinforced with additional references.
- Future Research Directions: The manuscript successfully identifies gaps in hydrogel research, particularly regarding safety, biocompatibility, and clinical translation. It would be helpful to discuss potential strategies for overcoming these challenges.

---

## Round 0.2 · Major Revisions

Dear authors,

Thanks for your revisions.

While I do understand that this is a narrative review, one crucial aspect that needs addressing really is "critical insights or comparative discussion." Future perspectives should also be discussed. Your work must offer insights or address a gap in literature - please, consider this carefully. See the reviewers' comments.

Reviewer 1 ·

Basic reporting

I have no further queries. The manuscript can now be accepted for publication.

Experimental design

Not applicable

Validity of the findings

Not aplicable

Additional comments

No

Reviewer 2 ·

Basic reporting

no comment

Experimental design

no comment

Validity of the findings

no comment

Additional comments

no comment

Reviewer 3 ·

Basic reporting

• The manuscript addresses a rapidly evolving field at the intersection of materials engineering, drug delivery, cancer therapeutics, and biomedical engineering. It holds good educational value, especially for early-career researchers entering this interdisciplinary domain.
• A significant number of references cited are from before 2020, while few citations reflect the advancements made between 2021 and 2024. The reference list should be updated to include recent and relevant studies to strengthen the review’s relevance and comprehensiveness.
• The inclusion of comparative analysis tables summarizing the different technologies and applications discussed, along with their advantages, limitations, and clinical outcomes, would greatly improve clarity. Additionally, the manuscript would benefit from figures illustrating key concepts and recent findings.

Experimental design

• The manuscript falls within the scope and aims of the journal. However, as a review, it lacks a clearly defined methodology, such as inclusion/exclusion criteria and details of the databases searched. This undermines the review’s transparency and reproducibility.
• The current narrative tends to summarize existing literature without offering critical insights or comparative discussion. A dedicated section evaluating the recent challenges and advances, along with future perspectives, should be incorporated, preferably merged with or placed in the conclusion.

Validity of the findings

N/A

---

## Round 0.3 · accepted · Accept

Dear authors,

All issues have now been addressed. I am happy to let you know that i am accepting your manuscript for publication. Thank you.

Reviewer 3 ·

Basic reporting

The authors addressed all comments raised by the reviewer and editor, and can be accepted now for publication in PeerJ.

Experimental design

No more comments

Validity of the findings

No more comments